# Occurrence of Biogenic Amines Producers in the Wastewater of the Dairy Industry

**DOI:** 10.3390/molecules25215143

**Published:** 2020-11-05

**Authors:** Petra Jančová, Vendula Pachlová, Erika Čechová, Karolína Cedidlová, Jana Šerá, Hana Pištěková, František Buňka, Leona Buňková

**Affiliations:** 1Department of Environmental Protection Engineering, Faculty of Technology, Tomas Bata University in Zlín, Vavrečkova 275, 760 01 Zlín, Czech Republic; e_cechova@utb.cz (E.Č.); k.cedidlova@seznam.cz (K.C.); sera@utb.cz (J.Š.); pistekova@utb.cz (H.P.); bunkova@utb.cz (L.B.); 2Department of Food Technology, Faculty of Technology, Tomas Bata University in Zlín, Vavrečkova 275, 760 01 Zlín, Czech Republic; pachlova@utb.cz; 3Food Research Laboratory, Department of Logistics, Faculty of Military Leadership, University of Defence, Kounicova 65, 662 10 Brno, Czech Republic; frantisek.bunka@gmail.com; 4University Institute, Tomas Bata University in Zlín, Nad Ovčírnou 3685, 760 01 Zlín, Czech Republic

**Keywords:** wastewater, bacteria, decarboxylase activity, biogenic amines

## Abstract

Out of six samples of wastewater produced in the dairy industry, taken in 2017 at various places of dairy operations, 86 bacterial strains showing decarboxylase activity were isolated. From the wastewater samples, the species of genera *Staphylococcus, Lactococcus, Enterococcus, Microbacterium, Kocuria, Acinetobacter, Pseudomonas, Aeromonas, Klebsiella* and *Enterobacter* were identified by the MALDI-TOF MS and biochemical methods. The *in vitro* produced quantity of eight biogenic amines (BAs) was detected by the HPLC/UV–Vis method. All the isolated bacteria were able to produce four to eight BAs. Tyramine, putrescine and cadaverine belonged to the most frequently produced BAs. Of the isolated bacteria, 41% were able to produce BAs in amounts >100 mg L^−1^. Therefore, wastewater embodies a potential vector of transmission of decarboxylase positive microorganisms, which should be taken into consideration in hazard analyses within foodstuff safety control. The parameters of this wastewater (contents of nitrites, nitrates, phosphates, and proteins) were also monitored.

## 1. Introduction

Milk and dairy products represent a suitable environment for the growth of microorganisms, which may influence, through their metabolic activity (positively or negatively), the product quality. Both Gram-positive and Gram-negative bacteria having decarboxylase activity may occur in milk and dairy products. These include some lactic acid bacteria (*Lactobacillus, Pediococcus, Streptococcus*) or, for instance, bacteria genera: *Bacillus, Clostridium, Pseudomonas, Staphylococcus, Micrococcus* [1,2,3,4,5,6,7].

Decarboxylases are lyase-class enzymes responsible for catalyzing the decarboxylation of amino acids to form biogenic amines (BAs). BAs are naturally occurring, low molecular weight organic nitrogen compounds; based on their chemical structures, they can be divided into several groups: aliphatic (putrescine, cadaverine, spermine and spermidine), aromatic (tyramine, phenylethylamine) or heterocyclic (histamine, tryptamine). Sometimes, putrescine, cadaverine, spermine, spermidine and agmatine are classified as polyamines [1]. These substances are indispensable for organisms (plants, animals, microorganisms) and they participate in a wide range of physiological processes, such as cell proliferation, growth, differentiation and neurotransmission. They also affect protein synthesis or biosynthesis of DNA and RNA [8,9]. On the other hand, they have negative effects on human health, because in excessive quantities they can show undesirable, or even toxic, effects [1]. In a wide range of dairy industry products (e.g., milk, cheese, curds, and yoghurt), BAs were detected in toxicologically not insignificant quantities [10,11].

The production of BAs by bacteria can be influenced by several external factors through affecting the kinetics of decarboxylase reactions, the growth of the aminogenic microorganisms and also the expression of their decarboxylase potential. BA synthesis in bacteria is related to energy gain or resistance to an acidic pH.

The production of BAs is dependent on intrinsic and extrinsic factors of the environment. In other words, the factors influencing the production of BAs not only include the presence of microorganisms with decarboxylase activity, but also the presence of precursors (amino acids), available sources of carbon (e.g., glucose) and suitable external conditions for the growth of microorganisms (temperature, pH value, NaCl concentration, aero-/anaerobiosis, growth phase of the cells and others) [12,13,14]. It has been proven that the production of BAs by bacteria is a property specific to certain strains but not to a given species [15].

The presence of microorganisms in wastewater produced in the dairy industry is caused on the one hand by flushing some process-important microorganisms out into the wastewater during processing, but also—on the other hand—due to the contamination occurring in these manufacturing processes. As shown by some pilot studies, BA producers are present even in these waters. However, it is not quite clear from the bibliography what importance this vector (wastewater) has for the transfer of decarboxylase-positive microorganisms, which may influence the level of food safety controls. Therefore, the objective of this study was to isolate and identify bacteria possessing decarboxylase activity (i.e., bacteria capable of producing BAs) not from milk or dairy products, but just from the dairy industry wastewater. Therefore, in order to support this microbiological study in terms of the occurrence of nutrients for contaminating microorganisms with decarboxylase activity, the parameters of this wastewater were also monitored (contents of nitrites, nitrates, phosphates and proteins). The presence of nutrient substances (containing nitrogen and phosphorus) has recently been connected with the increase in the eutrophication of surface waters. Moreover, the occurrence of nitrates in foods and in water is a serious threat to human health [16]. Finally, we evaluated the risks of the transfer of contaminating microorganisms with decarboxylase activity by the wastewater because other studies focused on this topic are not offered in the available bibliography at present. The final consequence (impact) of the acquired results will be considering this vector in the hazard analysis within the framework of the Hazard Analysis and Critical Control Points (HACCP) system. If the wastewater contains microorganisms with decarboxylase activity, there is a hazard of cross-contamination because transfer of the latter mentioned microorganisms from the wastewater to the production hall/line could occur, e.g., due to insufficient personal hygiene (personnel that could handle the wastewater and could also enter the production hall, e.g., service and maintaining personnel). Therefore, contamination of raw material and/or products could happen and the amounts of BAs in final products could be significantly increased.

## 2. Results

### 2.1. Identification of Bacteria Isolated from Wastewater Samples

In 2017, in two seasons (summer and autumn), wastewater samples (A–F, in triplicate) were obtained from six offtake points (the same sampling points in summer and autumn of one dairy) that are linked to the processing of various dairy products and are crucial for drainage of wastewater from the latter mentioned lines (Table 1). Firstly, data from both seasons (summer and autumn) were compared with each other and the results from summertime were not significantly different from the results from autumn time (*p* > 0.05). We used non-parametric Wilcoxon test independently for each sampling point. Therefore, the data were merged and presented as the results obtained in year 2017. Using the MALDI-TOF MS and biochemical methods, 77 bacterial strains belonging to 19 bacterial genera (see Table 2) out of 89 isolated colonies, plus three fungi/candida—*Candida parapsilosis* and *Metschnikowia pulcherrima* in the A sample, *Rhodotorula mucilaginosa* in the C sample—were successfully identified. Approximately 10% of the isolated microorganisms were not identified by the use of the MALDI-TOF MS and biochemical methods. Out of 77 isolated and identified bacterial strains (Table 1 and Table 2), 34 were Gram-positive, represented mainly by the genera of *Lactococcus* (35% of identified Gram-positive bacteria), *Staphylococcus* (21%), *Microbacterium* (12%), *Enterococcus* (9%), *Kocuria* (9%) and 43 Gram-negative mostly species of the genera *Acinetobacter* (35% of identified Gram-negative bacteria), *Pseudomonas* (19%), *Aeromonas* (14%), *Chryseobacterium* (9%), *Enterobacter* (7%) and *Klebsiella* (7%).

### 2.2. Bacterial Decarboxylase Activity, Determination of Biogenic Amines

In the case of individual isolated bacteria (86 bacterial strains), their decarboxylase activity was tested, i.e., the ability to produce BAs. Using the HPLC/UV–Vis method, the production of eight BAs was monitored, namely histamine (HIM), tyramine (TYM), phenylethylamine (PHM), tryptamine (TRM), putrescine (PUT), spermine (SPE), spermidine (SPD) and cadaverine (CAD).

*In vitro* production of the monitored BAs was noted in all tested bacterial strains. Individually isolated bacteria produced four to eight BAs in various combinations and quantities. Apart from the samples, the used decarboxylation medium (clean, without inoculated bacteria) was also analyzed and none of the monitored BAs were detected there.

Out of the samples from the dairy industry wastewater, from which the bacteria with decarboxylase activity were isolated, the highest share, among the produced and detected BAs, belonged to TYM, PUT and CAD. These BAs were produced by 42%, 37% and 32% tested bacterial genera in quantities >100 mg L^−1^, respectively. TYM in an amount >100 mg L^−1^ was produced by bacteria, among which the following representatives of the genera *Pseudomonas, Acinetobacter, Lactococcus, Kocuria, Enterococcus, Microbacterium, Staphylococcus* and *Exiguobacterium* were identified. The significant isolated PUT producers included representatives of the genera *Aeromonas, Enterobacter, Acinetobacter, Klebsiella, Lactococcus, Staphylococcus* and *Microbacterium*. CAD was produced in an amount >100 mg L^−1^ by the representatives of the genera *Klebsiella, Acinetobacter, Kocuria* and *Lactococcus* (see Table 2). TRM was produced in an amount >100 mg L^−1^ by 30% of the identified Gram-negative bacterial strains, where the representatives of the genera *Aeromonas, Klebsiella* and *Chryseobacterium* were among the significant producers. SPE was produced in an amount >100 mg L^−1^ only by the bacterium of the genus *Klebsiella* isolated from the C sample.

From the C sample (rinsing water from the production of Gervais-type cream cheese), we isolated *Klebsiella pneumoniae*, which produced four BAs (TRM, PUT, CAD and SPE) in an amount >100 mg L^−1^ (see Table 2). In addition, from the D sample (wastewater from the outdoor sewer), a bacterium capable of producing six BAs in an amount >100 mg L^−1^—TRM, PHM, PUT, CAD, HIM and TYM—was isolated. Unfortunately, this Gram-positive coccus (positive catalase, positive oxidase) could not be identified by the available methods.

### 2.3. Wastewater Monitored Parameters

In addition to microbiological investigations, other parameters of wastewater, such as nitrite, nitrate, phosphate concentrations and the protein contents, were monitored in all the obtained samples (see Table 3).

The nitrite concentrations in the wastewater samples (A–F) ranged from 0.1 to 1.5 mg L^−1^. The highest concentration of nitrates was found in the E sample (54.1 mg L^−1^), in other samples the nitrate concentration was below 50.0 mg L^−1^. The phosphate concentrations measured with the samples A to F fluctuated in the range from 0.8 to 662.4 mg L^−1^, where the highest concentration was found in the E sample (rinse water from a cheese-making unit with a share of whey). In the tested samples (A to F), concentrations of proteins were found in the E sample at 127.3 ± 1.6 mg L^−1^, in the A sample (processed cheese plant rinse water) at 97.9 ± 0.8 mg L^−1^ and the F sample (curd unit rinse water) at 37.0 ± 0.5 mg L^−1^.

## 3. Discussion

In the studies published by Shivsharan et al. [17,18], bacteria genera of *Lactobacillus, Pseudomonas, Arthrobacter, Microbacterium, Staphylococcus, Enterococcus* and others were isolated from dairy industry wastewater. Nevertheless, some other studies dealt with the bacteria isolation and identification from dairy industry wastewater [19,20], where they described the presence of bacteria genera of *Alcaligenes, Leuconostoc, Lactobacillus, Staphylococcus* or *Lactococcus*. In the aforementioned studies, the isolated and identified bacteria were of the same genera that we managed to isolate and identify. However, these studies were not concerned with the decarboxylase activity of these isolated bacteria. 

In milk and dairy products, BAs, or their producers, are generally present. According to the bibliography, the Gram-negative bacteria, mostly *Enterobacteriaceae* (e.g., *Escherichia coli, Hafnia alvei, Klebsiella pneumoniae, Morganella morganii* or *Serratia* sp.), and bacteria of the genus *Pseudomonas* present in milk are capable of producing HIM, PUT and CAD [21]. The production of PUT and CAD by the representatives of *Enterobacteriaceae* isolated from curd and cheese samples are described in the study by Torracca et al. [22]. However, lactic acid bacteria (LAB) are considered major producers of BAs in cheese, in particular representatives of the genera *Enterococcus, Lactobacillus, Leuconostoc, Lactococcus* and *Streptococcus* [23,24,25]. The LAB isolated from cheese are capable of producing TYM, PUT or CAD [26]. In the range of sour cream cultures, the production of HIM was described in addition to the production of TYM [27]. The results of our study (focused not on milk and dairy products but on dairy industry wastewater) point out that, even in this wastewater, significant BA producers (i.e., the bacteria with decarboxylase activity) are present. Among the bacteria isolated and identified by us, neither the *Enterobacteriaceae* species nor LAB were missing. These species are important producers of PUT, TYM, CAD and TRM in particular. The polyamines SPE and SPD frequently occur in raw cow milk [28], and SPD was not produced at a significant level by the bacteria isolated by us. In other words, more than 80% of all isolated bacteria did not produce SPD at all. In their study, Marino et al. [29] monitored the *in vitro* production of SPD by bacteria isolated from cheese; however, even they did not confirm the ability of the isolated enterobacteria (e.g., *Enterobacter, Serratia, Escherichia coli*) to produce this type of BA. Apart from this, they did not confirm the SPE production, either. In our study, SPE was produced *in vitro* by isolated bacteria in quantities from 20.7 to 79.9 mg L^−1^, almost by all isolated strains. 

In a comparison with the production of other BAs (particularly PUT, TYM and CAD), the production of HIM by the identified bacteria was significantly lower, it fluctuated within the range from not detected (ND—BA under detection limit) to 28.5 mg L^−1^. Similar quantities of produced HIM were measured by the authors of studies monitoring the decarboxylase activity of bacteria isolated from cheese [5,29].

The submitted study results show that the bacteria that normally occur in milk or dairy products (cheese, curd) are also present in the dairy industry wastewater and their decarboxylase activity (i.e., the ability to produce BAs) is not negligible—a lot of these bacteria were able to produce in vitro BAs in quantities of hundreds of mg L^−1^. It was evident from the results that the rinsing water contains both starter and non-starter lactic acid bacteria but also contaminating microorganisms. A not negligible share of isolated microorganisms produce BAs in abundant quantities. Therefore, wastewater represents a potential vector through which the decarboxylase-positive microorganisms may penetrate into the products as secondary contamination and thus influence the final product safety. Therefore, it is necessary to include this vector of contamination in the hazard analysis within the Hazard Analysis and Critical Control Points (HACCP) system, to determine its risk and adopt adequate control measures, probably at the level of an operational program of necessary preconditions.

In separate wastewater samples (samples A to F), the presence of monitored BAs was analyzed. They were not present here or they were under the detection threshold of the used analytical method. Gubartallah et al. [30] determined BAs in environmental water (namely seawater samples) and BAs were also not detected there.

Milk processing is usually considered to be the largest source of industrial wastewater, particularly in Europe [31]. The largest share of the dairy industry wastewater is represented by washing water produced in operations involving equipment washing, flushing out the production lines that are being transferred to another product type, putting in service, shutdowns and changes in operations. It is assumed that approximately 2% of the total volume of processed milk could get into the sewer systems [32]. According to the reference document on best available techniques, 2013 (BREF), in the dairy industry, the wastewater problem consists of relatively high values of contamination indicators (COD, BOD) as well as the presence of nutrient substances (containing nitrogen and phosphorus), the content of which in static and running waters has been recently connected with the increase in the eutrophication of surface waters. Therefore, in addition to microbiological investigations, other parameters of wastewater, such as nitrite, nitrate and phosphate concentrations, were monitored in all the obtained samples; in addition, the protein contents were also analyzed (Table 3). The aforementioned substances may serve as nutrients for microorganisms and they therefore support the possibility of the transfer of bacteria with decarboxylase activity in this way. The major legislative instrument of the European Community (EC) governing the discharge of nitrites into the aquatic environment is the Directive 2006/11/EC of the European Parliament and of the Council on Pollution, caused by certain dangerous substances discharged into the aquatic environment of the community (codified version). According to this directive, nitrites fall into list II, containing substances that have a deleterious effect on the aquatic environment, which can, however, be confined to a given area and which depend on the characteristics and location of the water into which such substances are discharged. In Decree No. 252/2004 Coll. (Czech Republic), the maximum permissible concentration of nitrites in drinking water is stipulated at 0.50 mg L^−1^. In two of the six offtake points (samples A to F), the nitrite concentrations were determined at lower levels than 0.50 mg L^−1^, which even met the requirements for drinking water. In the C sample, this concentration was 0.5 ± 0.1 mg L^−1^, while higher concentrations (<1.60 mg L^−1^) were determined in the D, E and F samples.

Similarly, nitrates are present in all water types. In Decree No. 252/2004 Coll. (Czech Republic), on hygienic requirements for drinking water, the maximum permissible concentration of nitrates is 50 mg L^−1^, but in water for infants the maximum permissible concentration is only 10 mg L^−1^. The highest concentration of nitrates was found in the E sample (rinse water from a cheese-making unit); however, in all other samples (A to D, F), the nitrate concentrations were rather low, even in compliance with the drinking water limits.

The maximum permissible concentration of phosphates for drinking water is stipulated at 3.5 mg L^−1^ (Decree No. 252/2004 Coll., Czech Republic). Substantially higher concentrations of phosphates were found in the tested samples from the dairy industry wastewater, which could have been expected due to the use of some processed cheese salts (mostly orthophosphates, diphosphates, triphosphates and polyphosphates) [33] or due to the higher concentration of phosphorus in whey (up to 1 g L^−1^ of phosphorus) [34]. 

As milk and whey contain a number of proteins [35], the protein concentrations were monitored in the tested samples from the dairy industry wastewater. In the tested samples (A to F), concentrations of proteins were found in three samples and ranged from 37.0 to 130.0 mg L^−1^. It is well known that proteins are among the contaminating organic substances and the fact that their concentrations in wastewater fluctuate, by orders, in tens of mg L^−1^, as documented by the study of Westgate and Park [36].

## 4. Materials and Methods

### 4.1. Wastewater Sample Characteristics

Samples from the dairy industry wastewater (A–F) were taken in one dairy in Czech Republic (Central Europe). In 2017, in two seasons (summer and autumn), wastewater samples were obtained from six offtake points (the same sampling points in summer and autumn) that are linked to the processing of various dairy products (Table 1). Samples of wastewater (A–F, in triplicate) were taken in sterile sample tubes and consequently, individual bacteria were isolated and identified from these samples of wastewater.

### 4.2. Isolation of Bacteria from the Wastewater Samples and Their Identification

The following cultivating media were used for the isolation of individual bacteria: meat peptone agar (HiMedia, Bombai, India) as a universal medium, identifying medium MRS (HiMedia) for the detection of the bacteria presence of the genus *Lactobacillus*, identifying medium M17 (HiMedia) for the milk coccobacilli collection, BHI (HiMedia) for the collection of more nutrient intensive bacteria, Endo’s medium (HiMedia) for the collection of enterobacteria, Sabouraud agar for the collection of yeasts and fungi and Slanetz–Bartley cultivating medium for the collection of enterococci.

For A and B wastewater samples, 1000 µL of each sample was always inoculated (due to a lower number of microorganisms there), while for C, D, E and F samples, 100 µL of decimal-diluted solutions of each was always inoculated for all seven medium types. The cultivation took place for 48 h at 30 °C, or 37 °C in case of enterobacteria and 25 °C on the Sabouraud agar for a period of five days.

The bacterial colonies of various phenotypes selected at random from Petri plates with countable numbers of colonies were purified by re-inoculation (three times) to individual colonies. This yielded pure colonies that were consequently identified by the use of commercially available sets of micro-tests (API 20 NE, API Staph API 20 Strep, bioMérieux, Marcy l’Etoile, France, ENTEROtest 24 from Pliva-Lachema Diagnostika, Brno, Czech Republic) according to the manufacturer’s instructions and by the use of other conventional phenotype tests (growth in the presence of various NaCl concentrations, growth at various temperatures, gas production and the catalase test). The results of the ENTEROtest 24, extended with regular tests, were evaluated by the use of the TNW software application (Pliva-Lachema Diagnostika) and the API results were evaluated by the tool for apiweb identification (bioMérieux).

The isolated bacteria were also identified by the MALDI-TOF MS method. Individual colonies were suspended in 150 µL of distilled water and 450 µL of ethanol. The samples prepared in this way were frozen at minus 25 °C and then analyzed using the MALDI-TOF MS instrument (Bruker Daltonics GmbH, Leipzig, Germany), by the earlier described method [37]. The obtained spectra were processed by the use of BioTyper software, version 2.0 (Bruker Daltonics).

### 4.3. Determination of Biogenic Amines

This study was focused on the monitoring of the production of the BAs and polyamines (HIM, TYM, PHM, TRM, PUT, SPE, SPD, CAD) by the bacteria isolated from the samples of dairy industry wastewater. Production of the above BAs was monitored in the cultivation broth after precolumn derivatization with dansyl chloride. Supernatant was diluted 1:1 (*v/v*) with 0.6 M perchloric acid (Acros, Geel, Belgium). Three independent extractions were performed on each culture sample. Subsequently, mixtures were derivatized using dansyl chloride (Sigma-Aldrich, St. Louis, MO, USA) with 1,7-heptanediamine (Fluka, Buchs, Switzerland) as an internal standard. The preparation of decarboxylation medium for the BAs production determination by the monitored bacteria was identical to the methodology described in the paper of Lorencová et al. [38]. The quantity of eight BAs was determined by the technique of liquid chromatography (Lab Alliance, State College, PA, USA and Agilent Technologies, Agilent, Paolo Alto, CA, USA). Chromatographic separation (ZORBAX Eclipse XDB-C18, 50 × 3.0 mm, 1.8 μm, Agilent Technologies) and spectrophotometric detection (λ = 254 nm) has already been described in the paper of Dadáková et al. [39]. The chromatograms were evaluated by the use of the Clarity software application.

### 4.4. Wastewater Monitored Parameters

The determination of nutrient concentrations, in particular of nitrogen and phosphorus, are among the key parameters normally monitored in wastewater. Therefore, the concentrations of nitrite, nitrates and phosphates were monitored in the samples obtained from the dairy industry wastewater (A to F samples), the content of proteins was determined in individual samples as well.

#### 4.4.1. Determination of Nitrites

The nitrite concentration was determined spectrophotometrically (by the Greiss technique) [40]. The nitrites present in the sample react with sulfanilic acid producing diazonium salts. The salts are consequently copulated with N-(1-Naphthyl) ethylenediamine (NED), yielding purple azo-dye suitable for photometric determination [41]. The samples (and the standards) were analyzed three times.

#### 4.4.2. Determination of Nitrates

The combined nitrate ion selective electrode [41] was used for the nitrate quantitative determination in the samples obtained from the dairy industry wastewater (A to F samples). The samples (and the standards) were analyzed three times.

#### 4.4.3. Determination of Phosphates

The phosphate concentration in the wastewater samples was determined by colorimetric analysis. In the presence of antimony ions, phosphates react in a medium of sulfuric acid with ammonium molybdate, yielding molybdophosphate acid. Consequently, the yellow complex is transferred by a reduction with ascorbic acid to a solution of phosphomolybdate blue [41,42]. The samples (and the standards) were analyzed three times.

#### 4.4.4. Determination of Proteins

In order to determine the protein content in the dairy industry wastewater samples, the spectrophotometric method according to Bradford was used [43]. The method is based on the interaction of proteins with Coomassie brilliant blue G-250 (CBB) in the acid medium. The samples (and the standards) were analyzed three times.

## 5. Conclusions

In this study, the *in vitro* production of eight BAs (HIM, TYM, PHM, TRP, PUT, SPE, SPD and CAD) by the bacteria isolated from samples of dairy industry wastewater in Central Europe taken in 2017 in various areas of dairy operations was monitored. Gram-positive bacteria, particularly lactococci, staphylococci and enterococci, were isolated from the wastewater samples. In the case of Gram-negative bacteria, representatives of the genera Pseudomonas, Acinetobacter, Enterobacter, Aeromonas and Klebsiella predominated. These bacteria were significant producers of TYM, PUT, CAD and TRM (>100 mg L^−1^).

The results of the study confirm the presence of BA producers (bacteria with significant decarboxylase activity) in the dairy industry wastewater. Therefore, this is a significant contaminating vector for intermediate and final products, which influences their safety. Due to this, it is necessary to include this vector of contamination in the implementation and maintenance of the hazard analysis within the HACCP system (e.g., in the area dealing with personal hygiene, especially for people that could enter both the area where wastewater is present and the production lines—notably service and maintenance personnel).

## Figures and Tables

**Table 1 molecules-25-05143-t001:** Wastewater samples (six offtake points) and identified bacteria.

Sample Type		Bacteria Present
A	production line of processed cheese and heat treated quark-type spreads	*Arthrobacter ilicis, Enterococcus casseliflavus, Lactococcus lactis, Leuconostoc mesenteroides, Lactobacillus* sp., unidentified gram-positive coccus; *Enterobacter asburiae, Enterobacter cloacae, Acinetobacter junii, Aeromonas caviae, Aeromonas eucrenophila, Pseudomonas pseudoalcaligenes, Pseudomonas gessardii, Pseudomonas brenneri, Pseudomonas fragi, Chryseobacterium scophthalmum*
B	rinsing water from forms for the production of natural cheeses	*Staphylococcus hominis, Staphylococcus carnosus,* unidentified gram-positive coccus; *Acinetobacter johnsonii, Acinetobacter radioresistens, Pseudomonas taetrolens*
C	production of cream soft cheese	*Enterococcus faecalis, Kocuria rhizophila, Lactococcus lactis, Lactococcus raffinolactis,* unidentified gram-positive bacillus; *Acinetobacter baumannii, Acinetobacter schindleri, Klebsiella pneumoniae, Klebsiella oxytoca, Raoultella ornithinolytica, Brevundimonas vesicularis, Pseudomonas graminis, Pseudomonas azotoformans*
D	outdoor drain water	*Microbacterium liquefaciens, Staphylococcus carnosus, Lactococcus lactis,* unidentified gram-positive cocci; *Kocuria rhizophila, Kocuria varians, Microbacterium mitrae, Exiguobacterium* sp., *Aeromonas veronii, Aeromonas caviae, Aeromonas media, Acinetobacter johnsonii, Acinetobacter schindleri, Acinetobacter lwoffii, Comamonas aquatica, Chryseobacterium scophthalmum, Pseudomonas fragi,* unidentified gram-negative bacilli
E	production of natural cheese	*Lactococcus lactis, Microbacterium lacticum, Lactobacillus* sp., unidentified gram-positive coccus; *Acinetobacter johnsonii, Enterobacter asburiae, Klebsiella oxytoca, Leclercia adecarboxylata, Chryseobacterium joostei*
F	rinsing water from quark production	*Microbacterium oxydans,* unidentified gram-positive coccus

**Table 2 molecules-25-05143-t002:** In vitro production of biogenic amines (minimum and maximum, mg L^−1^) by bacteria genera isolated from the dairy wastewater.

	N ^a^	TRM	PHM	PUT	CAD	HIM	TYM	SPD	SPE
*Acinetobacter* sp.	15	ND–73.1	ND–16.9	4.1–625.2	ND–277.6	ND–20.5	18.1–249.0	ND–33.8	27.8–79.9
*Aeromonas* sp.	6	ND–103.7	3.4–41.1	ND–1092.8	2.1–82.0	1.6–10.4	17.3–99.7	ND–14.6	21.3–61.3
*Brevundimonas* sp.	1	28.5 ± 0.2	8.1 ± 0.1	12.1 ± 0.1	6.5 ± 0.1	6.5 ± 0.1	42.1 ± 0.3	ND	55.8 ± 0.2
*Chryseobacterium* sp.	4	12.9–101.3	5.1–36.5	4.7–13.3	ND–170.3	ND–28.5	19.0–40.8	ND–3.1	29.2–55.8
*Comamonas* sp.	1	32.8 ± 0.3	4.0 ± 0.1	ND	7.2 ± 0.1	ND	37.0 ± 0.2	ND	56.0 ± 0.3
*Enterobacter* sp.	3	27.4–76.0	3.9–6.9	20.1–514.3	24.1–32.4	ND–7.6	24.7–33.0	ND–26.2	46.1–61.8
*Klebsiella* sp.	3	20.7–135.1	2.0–11.8	6.4–486.3	61.3–334.0	1.4–11.0	29.7–90.0	ND	47.6–244.5
*Leclercia* sp.	1	14.9 ± 0.2	7.0 ± 0.1	14.8 ± 0.3	7.1 ± 0.2	3.9 ± 0.2	20.9 ± 0.8	ND	47.1 ± 0.5
*Pseudomonas* sp.	8	ND–43.3	2.0–21.5	2.8–23.5	1.6–67.3	ND–11.2	10.2–701.2	ND	26.8–55.9
*Raoultella* sp.	1	48.6 ± 0.6	6.9 ± 0.2	13.6 ± 0.4	160.8 ± 3.1	6.1 ± 0.1	20.1 ± 0.3	ND	35.5 ± 0.3
unidentified gram-negative bacteria	2	51.5–53.6	3.7–7.3	190.5–792.7	5.7–68.2	2.6–6.5	25.1–36.7	ND	51.5–70.7
*Arthrobacter* sp.	1	32.5 ± 0.4	ND	ND	ND	3.4 ± 0.1	15.6 ± 0.4	ND	33.5 ± 0.7
*Enterococcus* sp.	3	29.3–40.4	7.2–19.9	19.4–33.6	ND–27.0	5.1–9.5	21.4–405.9	ND–2.5	34.6–49.3
*Exiguobacterium* sp.	1	64.0 ± 1.2	29.4 ± 0.6	23.7 ± 0.4	14.4 ± 0.3	16.7 ± 0.2	779.3 ± 15.1	ND	20.7 ± 1.1
*Kocuria* sp.	3	35.2–75.5	8.3–20.7	23.8–91.8	11.1–422.3	ND–13.3	44.2–327.3	ND	31.6–67.8
*Lactobacillus* sp.	2	ND–50.4	4.5–7.0	7.6–14.4	2.6–85.3	2.0–4.5	19.2–27.9	ND	37.0–42.9
*Lactococcus* sp.	12	ND–73.45	ND–15.1	2.5–1042.5	3.8–171.7	ND–14.3	11.9–515.5	ND–18.0	29.7–49.8
*Leuconostoc* sp.	1	ND	10.6 ± 0.3	4.7 ± 0.1	3.6 ± 0.1	2.0 ± 0.1	27.4 ± 1.5	ND	55.5 ± 2.3
*Microbacterium* sp.	4	17.9–69.4	5.9–22.5	4.95–275.5	ND–64.0	3.1–23.4	23.9–588.4	ND	28.6–51.5
*Staphylococcus* sp.	7	34.0–72.7	ND–27.8	7.4–1178.0	ND–94.8	ND–19.8	15.7–713.3	ND–23.1	23.9–64.8
unidentified gram-positive bacteria	7	39.8–462.4	ND–205.2	32.6–882.1	25.8–104.0	ND–107.8	20.4–570.5	ND–21.6	29.9–52.8

^a^ N—number of isolates of given bacteria group. ND—biogenic amines (BAs) under detection limit (0.3–1.4 mg L^−1^). BAs: tryptamine, TRM; phenylethylamine, PHM; putrescine, PUT; cadaverine, CAD; histamine, HIM; tyramine, TYM; spermidine, SPD; spermine, SPE.

**Table 3 molecules-25-05143-t003:** Wastewater monitored parameters [mg L^−1^].

Sample Type	Nitrite	Nitrate	Phosphates	Protein
A	0.1 ± 0.1	21.0 ± 0.4	21.0 ± 0.5	97.9 ± 0.8
B	ND	41.5 ± 0.5	0.8 ± 0.1	ND
C	0.5 ± 0.1	27.3 ± 0.4	37.0 ± 1.1	ND
D	1.5 ± 0.1	13.0 ± 0.5	25.2 ± 0.4	ND
E	1.0 ± 0.1	54.1 ± 1.2	662.4 ± 10.9	127.3 ± 1.6
F	1.0 ± 0.1	40.4 ± 0.6	43.8 ± 1.2	37.0 ± 0.5

ND—not detected.

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
