# Peer review of "Occurrence of Biogenic Amines Producers in the Wastewater of the Dairy Industry"

_molecules, 2020, doi:10.3390/molecules25215143_

Round 1

Reviewer 1 Report

The changes are enough.

Author Response

Reviewer 1 - Comments and Suggestions for Authors: The changes are enough.

Reviewer 2 Report

In the revised manuscript the authors followed the reviewer’s suggestions and were able, to some degree, improve their paper. However, I still have some concerns about the sampling process.

Further comments:

1 – Some abbreviations still require correction (e.g. lines 80, 313, 318).

2 – Regarding the sampling process:

  1. i) why did the authors decide to collect only 6 points from one factory? Why not perform the sampling in multiple factories and compare the results obtained among them?
  2. ii) The relevance of performing the sampling both in summer and autumn is not clear. Also, how can the authors justify that the differences between the results obtained in both seasons were not statistically significant?

3 – Table 2: The results presented in this table correspond exactly to what? Summer sampling? Autumn sampling? An average between the two seasons? How was the data statistically treated?

4 – Table 3 – Please correct the significant figures.

Author Response

Reviewer 2 - Comments and Suggestions for Authors:

In the revised manuscript the authors followed the reviewer’s suggestions and were able, to some degree, improve their paper. However, I still have some concerns about the sampling process.

Further comments:

1 – Some abbreviations still require correction (e.g. lines 80, 313, 318).

Information was corrected (specifically on lines: 35, 80, 275–276, 316–318, 323, 458).

2 – Regarding the sampling process:

  1. i) why did the authors decide to collect only 6 points from one factory? Why not perform the sampling in multiple factories and compare the results obtained among them?

Because this is a study that no one before us did, we chose one dairy and 6 sampling points in it (according to the accessibility of sampling points; places from the interior of the dairy /rinsing water/ and water obtained from the outdoor sewerage system are included).

Our goal was to determine whether seasonality (summer, autumn) has any significant effect on the results obtained (occurrence of individual microorganisms and wastewater parameters). We did not find any significant seasonal differences.

  1. ii) The relevance of performing the sampling both in summer and autumn is not clear. Also, how can the authors justify that the differences between the results obtained in both seasons were not statistically significant?

We elucidated the merging of data from summer and autumn seasons into a group of results from year 2017. For comparison, we used a non-parametric Wilcoxon test independently for each sampling point – see lines 85–89.

3 – Table 2: The results presented in this table correspond exactly to what? Summer sampling? Autumn sampling? An average between the two seasons? How was the data statistically treated?

The results presented in Table 2 corresponded to the whole year 2017 (the merged data set from the both summer and autumn sampling). For comparison purposes (the differences between strains and also species), Kruskal-Wallis and Wilcoxon tests were used (the differences were described only in the text).

4 – Table 3 – Please correct the significant figures.

Information (number of decimal points) was corrected.

Reviewer 3 Report

The manuscript entitled “Occurrence of Biogenic Amines Producers in the Wastewater of the Dairy Industry” presents results of the studies performed to identify the species of microorganisms able to produce biogenic amines, which were present in wastewater produced in the dairy industry.

As it was mentioned before, presented results were obtained with methods, which are dated and not representative for that type of analysis. In this regard, metagenome analysis should be performed obligatory. The authors confirmed that they are not able to obtain this type of data at their workplace. However, such analysis may be outsourced to other laboratories that have the required equipment.

Since Molecules is a journal with high IF value and publishes articles with high scientific background and significance, the manuscript in the current state should not be accepted. I do recommend its sending to other journals with IF ≤ 2.

Author Response

Reviewer 3 - Comments and Suggestions for Authors:

The manuscript entitled “Occurrence of Biogenic Amines Producers in the Wastewater of the Dairy Industry” presents results of the studies performed to identify the species of microorganisms able to produce biogenic amines, which were present in wastewater produced in the dairy industry.

As it was mentioned before, presented results were obtained with methods, which are dated and not representative for that type of analysis. In this regard, metagenome analysis should be performed obligatory. The authors confirmed that they are not able to obtain this type of data at their workplace. However, such analysis may be outsourced to other laboratories that have the required equipment.

The reviewer is right, of course, we are able to outsource the metagenome analysis. On the other hand, we do not understand why we should it make. We set our aim of the presented study and it is not clear how metagenome analysis help and improves the fulfillment of that goal. Please, clarify it to us in the context of our aim. According to our opinion, this type of analysis is interesting but not necessary and the association with our study is negligible.

Since Molecules is a journal with high IF value and publishes articles with high scientific background and significance, the manuscript in the current state should not be accepted. I do recommend its sending to other journals with IF ≤ 2.

We fully respect the quality of the journal „Molecules“, therefore, we decided to submit our valuable study to this honored and respected journal. We will be appreciated to have our work in this journal. We are looking forward to the citation of our work in this journal. Unfortunately, we do not understand how the possibility of publishing a manuscript in Molecules depends indispensably on metagenome analysis performance. Thank you for your elucidation.

Round 2

Reviewer 2 Report

The changes to Table 3 were not made properly. For the column "Nitrite" everything was well expressed except for Sample B, where the value read 0.01±0.00

Besides, avoid expressing error as ±0.0 as well in Table 2.

Author Response

Comments and Suggestions for Authors:

The changes to Table 3 were not made properly. For the column "Nitrite" everything was well expressed except for Sample B, where the value read 0.01±0.00

Besides, avoid expressing error as ±0.0 as well in Table 2.

The recommendation was accepted (Table 2 and 3).

Reviewer 3 Report

(no comment)

This manuscript is a resubmission of an earlier submission. The following is a list of the peer review reports and author responses from that submission.

Round 1

Reviewer 1 Report

The authors describe the results of decarboxylase activity of bacteria capable of producing biogenic amines of from the dairy industry wastewater to evaluate the risks of the transfer of contaminating microorganisms with the decarboxylase activity by the wastewater.

The results are very interesting because deal with an important environmental topic and because few studies focused on this topic. However, the final impact of the acquired results in the hazard analysis within the framework of the HACCP system is not clear.

I think that the manuscript can be improved with some items indicated below. Some papers could be cited to improve the introduction:

To improve the introduction add and discuss these references:

  1. Determination of nitrate and nitrite levels in infant foods marketed in Southern Italy ML Cortesi, L Vollano, MF Peruzy, R Marrone, R Mercogliano CyTA-Journal of Food 13 (4), 629-634
  2. Biogenic amines profile in processed bluefin tuna (Thunnus thynnus) products R Mercogliano, A De Felice, ML Cortesi, N Murru, R Marrone, A Anastasio CyTA-Journal of Food 11 (2), 101-107

Line 74 “taken in Central Europe, in 2017” more details are needed, where? during 2017 samplings were carried out in different seasons?

There is a problem regarding sampling, please describe the procedure and explain the small sample size. Has a statistical analysis been performed?

Line 121 “nitrogen and phosphorus”, explain why only these parameters.

Explain better why include wastewater vector of contamination into the hazard analysis within the HACCP system.

Check English and References section.

Reviewer 2 Report

In the manuscript entitled “Occurrence of biogenic amines producers in the wastewater of the dairy industry” the authors describe the analysis of six wastewater samples from the dairy industry and the identification of several bacteria capable of producing biogenic amines. From an overall perspective, I believe the novelty of the work is limited. Furthermore, the manuscript requires an extensive revision in its conceptualization, information, and discussion.

Further comments:

  1. The manuscript requires language editing. Some sentences are unclear (e.g. lines 34-35, 274-275) and often paragraphs are too long and confusing (e.g. 40-45). Authors are advised to either proofread the manuscript from a native English speaker or contact a language editing service provider.
  2. Authors introduce different abbreviations in the text (e.g. biogenic amines (BAs); putrescine (PUT), etc.) but often switch between the use of the full name and the abbreviation. Please use the full name at first instance and abbreviation afterward.
  3. There are some errors in different standard units throughout the manuscript. Such as for temperature as 30°C (should be 30 °C), µl (should be µL) among others. Authors should uniform units and nomenclatures.
  4. The introduction should be clearer and objective. It is not clear the importance of bacteria present in wastewater and the correlation the authors try to make to the dairy product quality. It would be mandatory to describe how the wastewater containing the bacteria could affect the food safety of products being produced.
  5. The description of the sampling process omits important and relevant information, as in which country the sampling occurred, why there were only six samples, were these samples all from the same factory, were from different factories, how many points were collected inside the same factory, were all in the same day or in different days.
  6. Authors should avoid using sentences like “according to our assumption of a lower number of microorganisms there” without providing a suitable explanation for the assumption.
  7. The methods used could use more information. Although the authors refer to past publications, a short description of the methodology could be provided, such as in the determination of biogenic amines. Especially when critical parameters of the methodology and of the method are discussed in the results (e.g. LOD).
  8. In Section 3.3 the authors should provide the complete full set of results for the different wastewater samples (A-F) in a Table, as it would simplify the interpretation and discussion of the results.
  9. The results obtained for the polyamines spermine and spermidine could be more critically discussed.

Reviewer 3 Report

The manuscript entitled “Occurrence of Biogenic Amines Producers in the Wastewater of the Dairy Industry” presents results of the studies performed to identify the species of microorganisms able to produce biogenic amines, which were present in wastewater produced in the dairy industry.

Major point is that the plate method used by the Authors for microorganisms isolation from wastes is not suitable in such an analysis. Despite the fact that it is a bit dated, it also allows us to isolate only these microorganisms which can be cultured in the laboratory. In an environment like wastes, there are present many other microorganisms, which can not be cultured in laboratory conditions (non-cultivated microorganisms), as well as other microorganisms that were not active during the isolation procedure. Therefore for such trials (analyses of organisms from wastewater), the genetic test of the microbiome is now mandatory. On the other hand, the MALDI TOF  analysis was performed to identify some species – this is a new and very useful technique. But still, presented MALDI-TOF identification was performed only for these species, which were isolated from wastewater and grown on the media in the laboratory. Therefore, due to the presence of microorganisms in the studied environment, which were not isolated during laboratory culturing or were present not-active forms, there should be performed metagenome analysis. It will give the proper identification of species present in dairy wastewater.

Authors also presented a screening of microorganisms only according to their ability for biogenic amines production, whereas the metabolome analysis of the entire metabolic profile should be performed.

Minor comments: in the introduction part there should be presented with more detailed biological activity of biogenic amines; the structure of the article should be changed (part with methods should be presented after results).

In summary, the data presented in the manuscript does not present valuable scientific significance and should be enriched with further analysis, which are mentioned above.